# Epidemiological investigation and analysis of the genetic evolution of duck circovirus in China, 2022

Peidong Li[1], Fuyou Zhang[1], Chunyang Bao[1], Hongmei Liu[1], Kai Yu[1], Hao Zhu[1], Xue Wang[1], Ke Shen[1], Tianyao Yang[2], Qingqing Song[1], Zhaoyang Li[2], Chunguo Liu[1]*

1 Group Biological Products R & D Center, Shandong Sinder Technology Co., Ltd, Qingdao, Shandong, China, 2 Shandong Sinder Technology Co., Ltd., Qingdao, Shandong, China

* liuchunguo@sinder.cn

## Abstract

Duck circovirus (DuCV) infection is an immunosuppressive disease that affects ducks and causes severe damage to their immune system. To elucidate the epidemiological characteristics of DuCV infection in China, a total of 2944 waterfowl samples were collected from 17 provinces from January to October 2022, and 612 DuCV-positive samples were identified. A descriptive statistical analysis was subsequently conducted. Furthermore, 51 near-full-length DuCV genome sequences were obtained, and molecular genetic evolution and recombinant analyses were performed. Geographically, Fujian Province had the highest rate of DuCV positivity (54.8%), followed by the Guangxi Zhuang Autonomous Region (30.4%). The rate of DuCV positivity was highest in samples from 21–40-day-old ducklings, accounting for 66.5% of the total positive samples. The most common pathogen involved in mixed infections with DuCV was parvovirus or *Riemerella anatipestifer*. Genetic and evolutionary analyses of the full genome sequences of 51 DuCV strains revealed that DuCV-1b and DuCV-2c were the most prevalent strains in China. Genetic recombination analysis suggested that the major parental sequences involved in the recombination of DuCV strains in ducks are present in Anhui, Sichuan, Shandong, and Guangxi. In addition, DuCV recombination events have occurred between strains with different genotypes or strains isolated from different countries. In summary, the DuCV epidemic in China is complex. There are two main co-circulating genotypes, those of the DuCV-1b and DuCV-2c strains, and coinfection of DuCV with other pathogens is a very common phenomenon in clinical practice. There is an urgent demand for vaccines against DuCV, and the protective efficacy of these vaccines against different DuCV genotypes needs to be carefully evaluated.

**Data availability statement:** The data presented in this study are openly available in GenBank under accession numbers OR387724 and OR387774. See S1 Table for the sample information.

**Funding:** This study was supported by the Shandong Sinder Technology Co., Ltd. (Project Number: JC20230313001). The funders had no role in study design, data collection and analysis, decision to publish, or preparation of the manuscript.

**Competing interests:** The authors have declared that no competing interests exist.

## Introduction

Circovirus is a single-stranded circular DNA virus with a diameter of 15–16 nm. Its structure consists of a nucleocapsid protein without an envelope [1,2]. Although viruses in the *Circoviridae* family can infect a wide range of vertebrates, they have a high degree of host specificity, and cross-species infection is limited to related species [3]. The known *Circoviridae* are divided into two genera, *Cyclovirus* and *Circovirus*. There are numerous members of the *Circovirus* genus, including human circovirus, porcine circovirus, pigeon circovirus, duck circovirus (DuCV), and finch circovirus [4,5] (https://ictv.global/taxonomy/). DuCV in particular is generally classified into three genotypes: DuCV-1, DuCV-2, and DuCV-3, the latter of which was recently identified and discovered in Hunan Province, China [6]. An additional genotype, DuCV-VS, based on DuCV-1, has been reported in the velvet scoter [3]. The DuCV genome contains three main open reading frames (ORFs): ORF1, encoding a viral replication-related protein; ORF2, encoding an immunogenic capsid protein (Cap); and ORF3, encoding an active apoptotic protein [2,7,8].

DuCV infection was first identified in Germany [9]. The disease then circulated worldwide and has been reported in Hungary, the United States, South Korea, Poland, China, Vietnam and many other countries [3,10–14]. Like other circoviruses, DuCV infection is characterized by severe immunosuppression [13]. In addition, infection with the virus can cause ducklings to grow slowly and display disorganized feathers [13,15], especially when they are coinfected with parvovirus. Wang et al. reported that DuCV infection causes severe damage to the immune system and that the immunosuppression caused by DuCV is key to secondary bacterial or viral infection [16]. In addition, mixed infection with a novel goose parvovirus and DuCV demonstrated a synergistic effect on virus replication and pathogenicity in ducks [17]. Moreover, DuCV infection also induces primary cholecystitis [18]. Importantly, DuCV infection is characterized by systemic infection, persistent infection, and horizontal and vertical transmission [19–21]; unfortunately, there is no commercial inactivated vaccine for DuCV infection because of a lack of a good culture system for isolating the virus [22]. These factors collectively may provide an opportune environment for the virus to circulate in the duck population and increase its infectivity and pathogenicity with other viruses or bacteria, ultimately resulting in great economic losses for farmers.

In this study, we analyzed the data of 612 DuCV-positive samples collected in China from January to October 2022 and conducted genetic evolution and homologous recombination analyses on all 51 obtained genome sequences. The goal of this study was to elucidate the pattern of DuCV infection in China and provide an epidemiological foundation for researching and developing DuCV vaccines.

## Results

### DuCV detection and descriptive analysis

A total of 2944 duck samples were collected from 17 provinces from January to October 2022 (Fig 1). Among these, 612 samples were positive for DuCV, with a positive rate of 20.8%; this value differed greatly among provinces, with a range from 0% to 54.8%. In

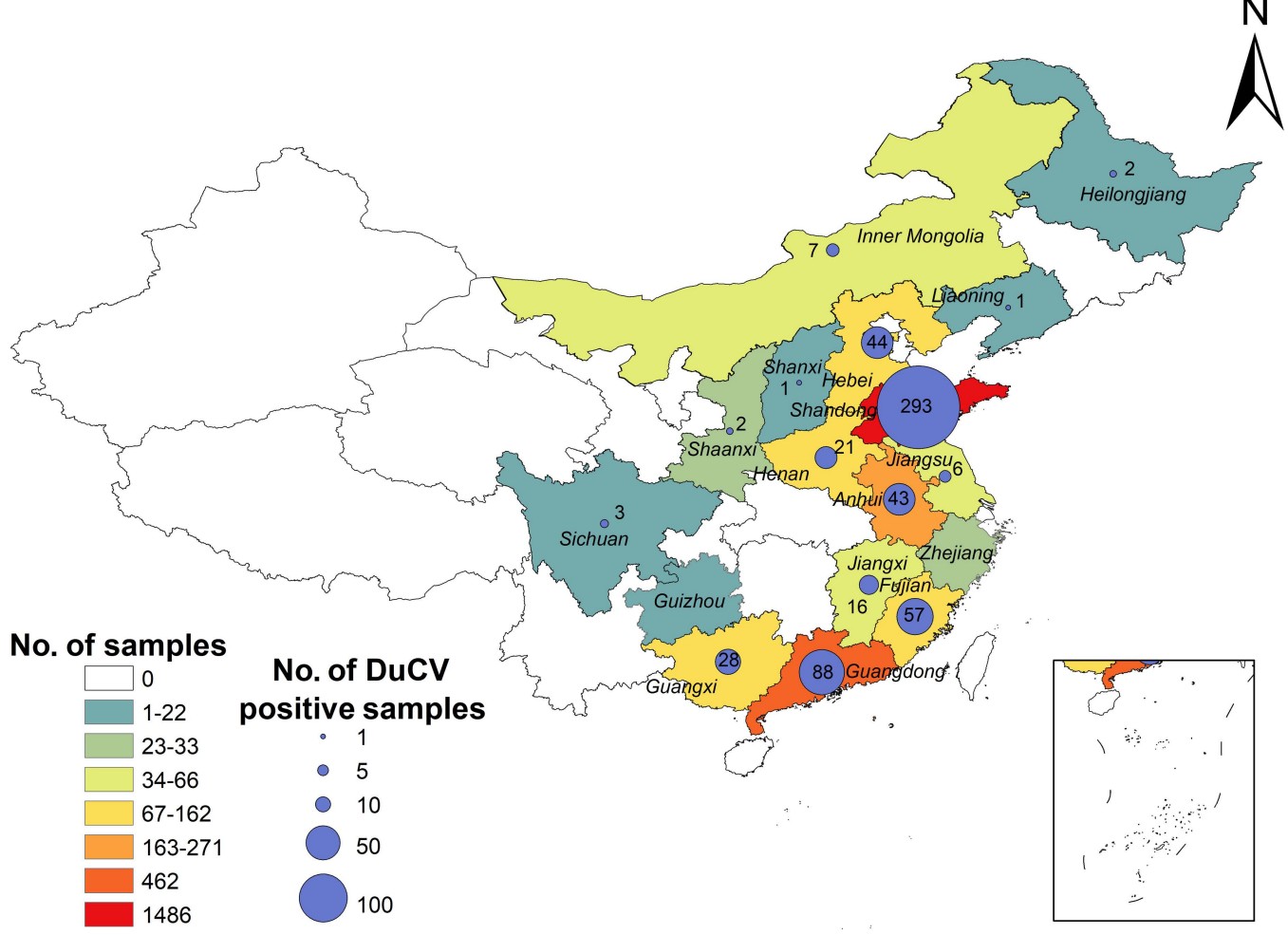

**Fig 1. Sample sources and number of DuCV-positive samples.** Different background colors represent different numbers of sample sources; the size of th e circles indicates the number of DuCV-positive samples, with specific quantities marked in the graph. The boundary data for China's provincial-level administrative divisions were obtained from the Resource and Environment Science Data Platform (https://www.resdc.cn/).

particular, Fujian Province had the highest rate of DuCV positivity (54.8% [57/104]), followed by the Guangxi Zhuang Autonomous Region (30.4% [28/92]), whereas Guizhou Province and Zhejiang Province were both negative for DuCV (Fig 1).

There was no obvious temporal pattern to the DuCV infections (Fig 2A). In this study, the greatest number of DuCV-positive ducks were aged 21–40 days, accounting for 66.5% of the population (Fig 2B). In addition, we detected one case (0.2%) of DuCV infection in a duck embryo. Some amount of DuCV was also detected in more than 100-day-old ducks (2.3%), while the oldest DuCV-infected ducks were 530 days old (Fig 2B).

Coinfection analysis revealed that among the 612 positive samples collected in this study, 207 had mixed infections, accounting for 33.8%. Among the DuCV-infected samples, parvovirus (including novel goose parvovirus and Muscovy duck parvovirus) and *Riemerella anatipestifer* were the main coinfective pathogens (Fig 3). Other pathogens, including duck astrovirus, duck hepatitis virus, duck reovirus, novel duck reovirus, avian influenza virus, fowl adenovirus, and *Pasteurella multocida*, were also detected in the DuCV-positive samples. Notably, coinfection with two or multiple pathogens, including bacteria and/or viruses, is a common phenomenon in clinical practice. We must also note that not all samples were tested for all pathogens, so the real coinfection rate may be higher than that shown in these data.

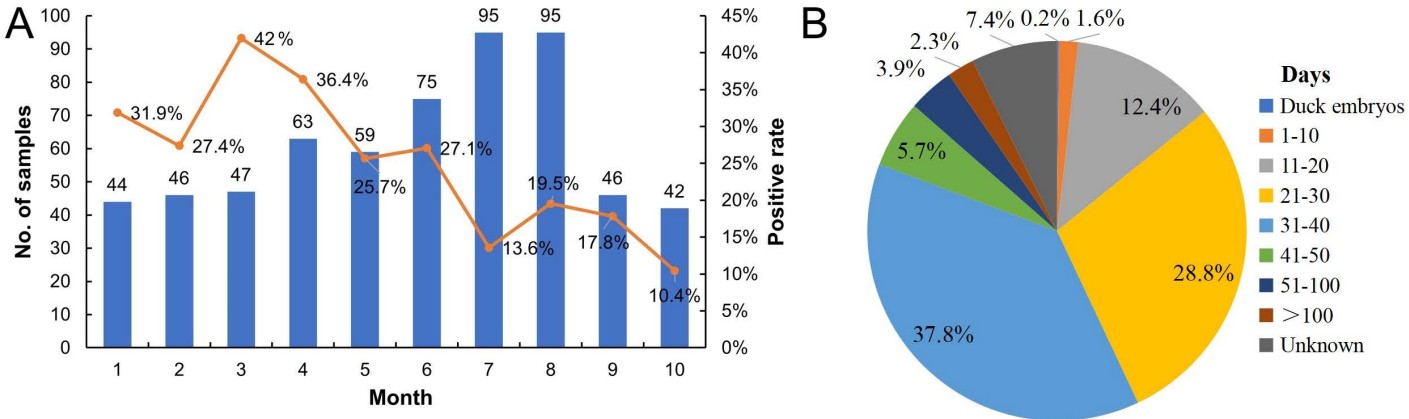

**Fig 2. DuCV detected from January to October 2022.** (A) Number and percentage of DuCV-positive samples identified each month from January to October; (B) Distribution of the number of DuCV-positive samples relative to the number of day-old DuCV-infected ducks.

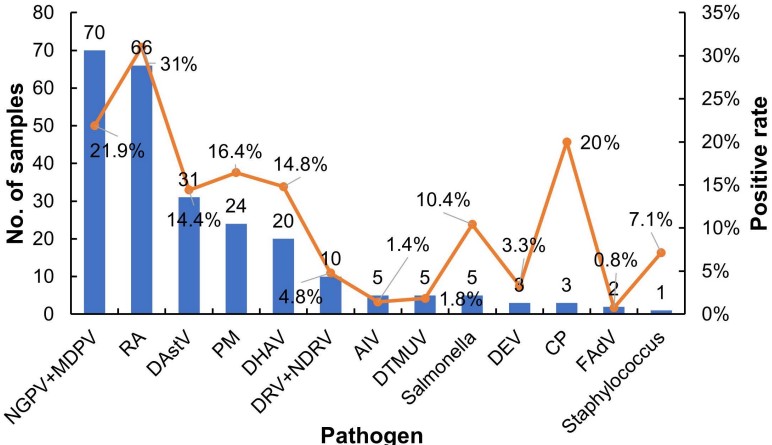

**Fig 3. Numbers and positive rates of mixed infection with DuCV and other viruses or bacteria in ducks. NGPV**: novel goose parvovirus, **MDPV**: Muscovy duck parvovirus, **RA**: *Riemerella anatipestifer*, **DAstV**: duck astrovirus, **PM**: *Pasteurella multocida*, **DHAV**: duck hepatitis A virus, **DRV**: duck reovirus, **NDRV**: novel duck reovirus, **AIV**: avian influenza virus, **DTMUV**: duck Tembusu virus, **DEV**: duck enteritis virus, **CP**: *Clostridium perfringens*, **FAdV**: fowl adenovirus.

## Genetic evolutionary analysis

In the present study, a total of 51 nearly complete genomic sequences of DuCV from eight provinces—Shandong, Anhui, Guangdong, Guangxi, Fujian, Jiangxi, Henan, and Sichuan—were obtained from DuCV-positive samples. These sequences were submitted to GenBank with accession numbers OR387724–OR387774; detailed information is listed in S1 Table. Like those of other DuCVs, the 51 genomic sequences of DuCV contained 3 ORFs. The results of phylogenetic analysis indicated that all the DuCV-1 and DuCV-2 genotypes existed in China before 2015, but after 2021, DuCV-1b and DuCV-2c were the main circulating genotypes, whereas the other DuCV genotypes gradually became less commonly encountered or even disappeared (Fig 4). According to the sequencing data obtained in this study, DuCV-2c was mainly found in southern China, DuCV-1b was detected in both southern and northern provinces of China, and DuCV-1c was detected only in Guangdong Province, in which multiple other DuCV genotypes co-circulated.

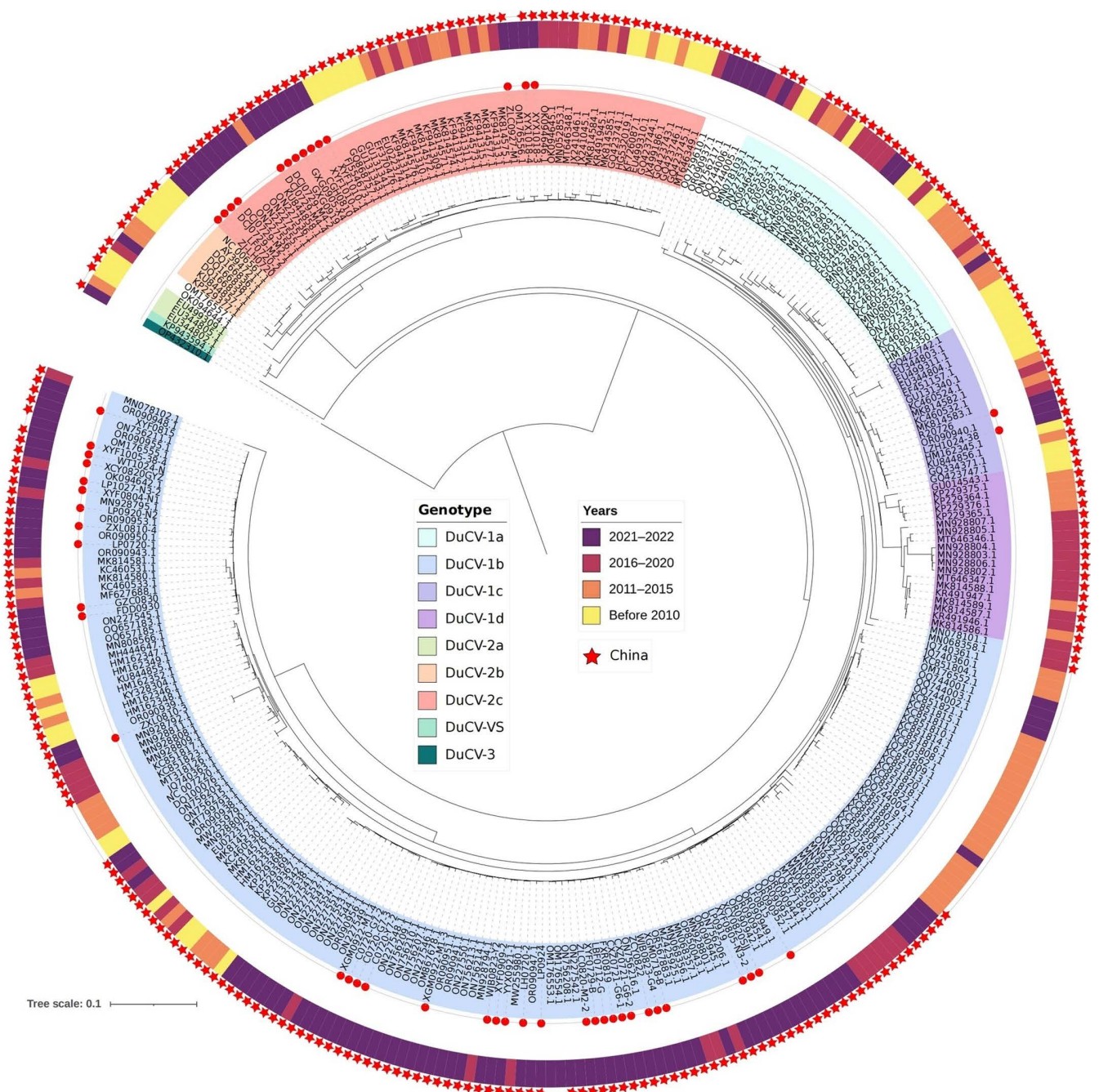

**Fig 4. DuCV genetic evolutionary tree based on the full-length genome.** All full-length gene sequences of DuCV prior to 2023 were downloaded from NCBI, and a phylogenetic tree was constructed with the 51 DuCV sequences identified in this study. The different colors in the inner circles represent different genotypes. The different colors in the outer circles represent different years. The red dots represent the DuCVs obtained in the present study. The red five-pointed stars represent the DuCV strains detected in China in different years.

In the present study, the nucleotide and amino acid sequences of the whole coding regions of all 51 DuCV ORF1/ORF2/ORF3 sequences were analyzed (Table 1). The nucleotide sequence identity was 93%–100% for the DuCV-1 strains and 97.5%–100% for the DuCV-2 strains; the ORF1 identity was 96.8%–100% for the DuCV-1 strains and 97.1%–100% for the

**Table 1. Identity or distance of nucleotide and amino acid sequences within and between different genotypes in the present study.**

| Comparison | genome | ORF1 | | ORF2 | | ORF3 | |
|---|---|---|---|---|---|---|---|
| | nt[a] | nt | aa[b] | nt | aa | nt | aa |
| Within DuCV-1 | 93%–100% | 96.8%–100% | 97.3%–100% | 90.6%–100% | 94.5%–100% | 97.4%–100% | 92.3%–100% |
| Within DuCV-2 | 97.5%–100% | 97.1%–100% | 99.3%–100% | 97.1%–100% | 97.7%–100% | 99%–100% | 97%–100% |
| Between DuCV-1 and DuCV-2 | 0.189 | 0.072 | 0.057 | 0.25 | 0.133 | 0.105 | 0.228 |

[a]Nucleotide sequence

[b]Amino acid sequence

DuCV-2 strains; the ORF2 identity was 90.6%–100% for the DuCV-1 strains and 97.1%–100% for the DuCV-2 strains; the ORF3 identity was 97.4%–100% for the DuCV-1 strains and 99%–100% for the DuCV-2 strains; and the average distances of the whole genome and the ORF1, ORF2, and ORF3 sequences between the DuCV1 strains and the DuCV-2 strains were 0.189, 0.072, 0.25, and 0.105, respectively. Amino acid sequence analysis revealed that the identity of ORF1 was 97.3%–100% for the DuCV-1 strains and 99.3%–100% for the DuCV-2 strains; the identity of ORF2 was 94.5%–100% for the DuCV-1 strains and 97.7%–100% for the DuCV-2 strains; the identity of ORF3 was 92.3%–100% for the DuCV-1 strains and 97%–100% for the DuCV-2 strains; and the average distances of the ORF1, ORF2, and ORF3 sequences between the DuCV1 strains and the DuCV-2 strains were 0.057, 0.133 and 0.228, respectively. The identity of the amino acid sequence of ORF2 was lower than that of ORF1, both within and between the DuCV-1 and DuCV-2 strains.

## Amino acid site analysis

Protein prediction analysis was performed on the basis of the deduced amino acid sequences of ORF1, ORF2, and ORF3 for the 51 DuCV strains. Compared with the first discovered reference strain, AY228555.1, all the isolates had an L16I mutation in a replication-related protein. F23L and K64R were replaced in all the DuCV-2c Cap proteins, and G18A mutations occurred in 50% (7/14) of the DuCV-2c Cap proteins (Fig 5); these mutations were located in potential B-cell epitopes (S1 Fig) and not in the 6 major mutation regions (amino acid residues: 3–15, 31–63, 104–124, 143–159, 177–213, and 232–238) [21]. The results of the tertiary structure prediction of the Cap protein suggested that these major mutation areas were mostly located outside the protein (Fig 7A–B). In addition, we observed that in DuCV-1b, when amino acids 47H, 183I, 197H, and 205K appeared simultaneously, amino acids 55N, 82Q, 106N, 107K, 194T, and 236D did not appear, and vice versa; analysis of the other DuCV-1b strains uploaded to GenBank further confirmed this phenomenon (Fig 5). When amino acids 47H, 183I, 197H, and 205K or amino acids 55N, 82Q, 106N, 107K, 194T, and 236D coexisted at the same time, the viruses carrying these two amino acid combinations formed two different clusters (Fig 6). The sites formed from these amino acids, some of which are located on the surface of the protein (Fig 7C), may affect virus glycosylation or phosphorylation (Table 2).

## Recombination analysis

Recombination analysis was performed using RPD4 software on the full-length genomic sequences of 51 DuCVs and reference strains available on the National Center for Biotechnology Information (NCBI) database prior to 2023. Seven distinct detection methods (RDP, GENECONV, bootscan, maximum chi square, Chimera, SISCAN, and distance plot) [23] were used in this study. The Bonferroni correction $p$ value was 0.05 for all analyses. Among the 317 DuCV gene sequences, a total of 23 recombination events occurred (as detected by 3 or more methods). The LP0920-N2, FDD0930, C0720-Y7-2, and ZCY0822 sequences in this study were identified as major parents; these strains were identified in samples from Anhui, Sichuan, Shandong and Guangxi, and the ducks in these four regions played important roles in the evolution of DuCV (Table 3, and S1 Table). Different DuCV genotypes were involved in all recombination events. For

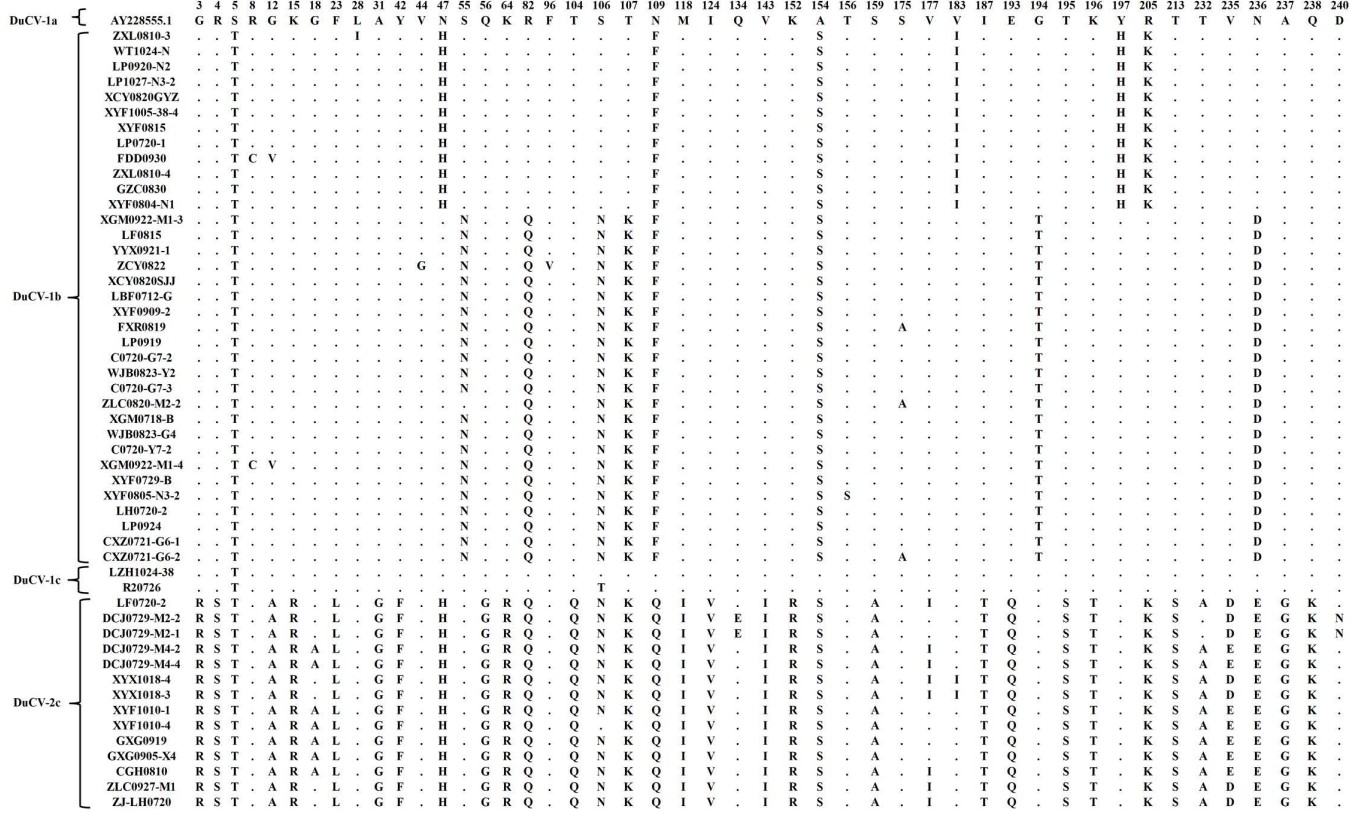

**Fig 5. Mutation of amino acids in OPF2 was detected in all the sequenced strains in this study.** The top numbers represent amino acid positions. The dots in the figure represent the same amino acids as those in reference strain AY228555.1.

**Table 2. Glycosylation and phosphorylation of some amino acid sites in the Cap protein of DuCV-1b.**

| Modification | Amino acid | | | | | | | | | | | | | | | | | |
|---|---|---|---|---|---|---|---|---|---|---|---|---|---|---|---|---|---|---|
| | 47 | 55 | | 82 | 106 | | 107 | | 183 | | 194 | | 197 | | | 205 | 236 | |
| | H | N | N | S | Q | R | N | S | K | T | I | V | T | | G | H | Y | K | R | D | N |
| N-linked glycosSylation | – | – | + NQT | – | – | – | – | – | – | – | – | – | – | – | – | – | – | – | – | – |
| Phosphorylation Kinase | – | – | – | + DNAPK | – | – | – | – | + PKC | – | + cdk5 | – | – | + PKC, unsp | – | – | + INSR, unsp | – | – | – | – |

The predictive analyses were performed with the reference sequences LP0920-N2 and LH0720–2.

"+" indicates the potential presence of a modifier site, whereas "–" indicates its absence.

DNAPK: DNA-dependent protein kinase, PKC: protein kinase C, cdk5: cyclin-dependent kinase 5, INSR: insulin receptor, unsp: nonspecific kinase prediction results.

example, recombination event 1 potentially involved recombination of the LP0920-N2 sequence of the DuCV-1b isoform with the ON227536.1 sequence of the DuCV-2c isoform (Fig 8A and 9A–C), whereas recombination event 7 may have involved recombination of the OQ657186.1 sequence of the DuCV-1b isoform with the AY228555.1 sequence of the DuCV-1a isoform (Fig 8B and 9D–F) as well as recombination of the German sequence AY228555.1 with the Chinese sequence OQ657186.1 (Fig 8B and 9D–F).

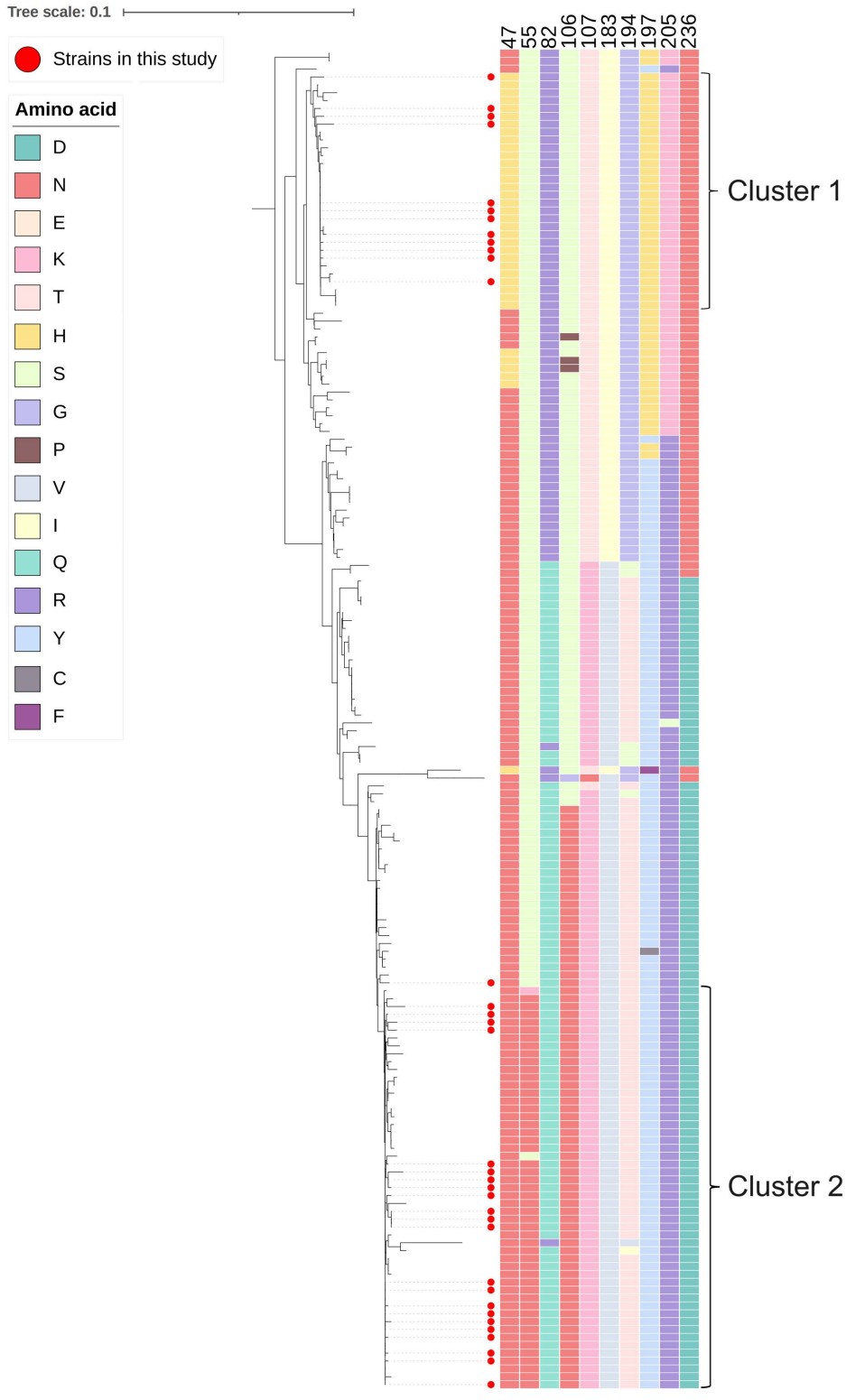

**Fig 6. ORF2 genetic evolution analysis of DuCV-1b.** The external ribbon represents amino acids at different positions.

**Table 3. Partial recombination events detected in the present study.**

| Event No.[a] | Recombinant sequence (Genotype) | Major parent[b] (Genotype) | Minor parent[b] (Genotype) | Evidence[c] |
|---|---|---|---|---|
| 1 | MT646347.1 (1d) | LP0920-N2 (1b) | ON227536.1 (2c) | R/G/B/M/C/S/T |
| 2 | LP0920-N2 (1b) | FDD0930 (1b) | EU344806.1[d] (1a) | R/G/M/S/T |
| 3 | ON227540.1 (1b) | C0720-Y7-2 (1b) | OK094643.1[d] (1a) | M/C/T |
| 4 | OR090946.1 (1b)<br>YYX0921–1 (1b)[e]<br>XYF0805-N3-2 (1b)[e] | ZCY0822 (1b) | OQ657183.1[d] (1b) | M/C/T |
| 5 | XGM0922-M1-4 (1b) | ON756210.1 (1b) | KP229366.1[d] (1a) | G/S/T |
| 6 | LP0720–1 (1b)<br>LP1027-N3-2 (1b)<br>FDD0930 (1b)<br>WY1024-N (1b)<br>GZ0830 (1b)<br>XYF0804-N1 (1b) | KC851823.1 (1b) | OQ657186.1[d] (1b) | M/S/T |
| 7 | MN068358.1 (1b) | OQ657186.1 (1b) | AY228555.1 (1a) | R/M/C/S |

[a]Only recombination events relevant to this study were counted.

[b]"Major" and "Minor" parents refer to parent sequences contributing the smaller and larger fractions of the recombinant sequence, respectively.

[c]R/G/B/M/C/S/T: RDP, GENECONV, bootscan, maximum chi-square, Chimera, SISCAN, and Distance Plot, respectively.

[d]Sequence used to infer unknown parents.

[e]Sequence with partial evidence of the same recombination event.

## Discussion

Livestock and poultry production accounts for nearly half of the global agricultural economy and is an important source of daily protein for humans [24]; among the meats of different fowl, duck meat is considered a healthy food [25]. According to data from the World Food and Agriculture Organization, the amount of duck meat produced in China was the highest in the world in 2021 (https://www.fao.org/faostat/zh/#data/QCL). However, compared with large-scale chicken farming, the management and hygiene of duck farming in China are suboptimal. A total of 2944 waterfowl samples were collected from 17 provinces from January to October 2022, and 612 DuCV-positive samples were identified. The positive rate differed greatly among the provinces, among which Fujian Province had the highest value (54.8%). DuCV infection affects mainly young ducks [26]. According to a study by Wan et al., ducks aged 40–60 days are susceptible to DuCV in southern China [27], but in the present study, most DuCV-positive ducks (66.5%) were aged 21–40 days, younger than the age reported by Wan et al. The main reason for this difference may be that the breeding period of meat ducks in China is approximately 40 days, and the sample size of meat ducks accounted for a large proportion of the current sample. The oldest duck infected DuCVs were 530 days old, suggesting that although young ducks are most susceptible to DuCV, older ducks can also be infected with the virus. Notably, we also detected DuCV in duck embryos, which indirectly confirmed that DuCV can infect ducks through vertical transmission. In addition, DuCV infection is an immunosuppressive disease; once ducks are infected with DuCV, secondary infections with other pathogens are not uncommon [20]. Among the DuCV-positive samples, many (33.8%) were also positive for other pathogens, among which parvovirus and *Riemerella anatipestifer* were the most common. Liu et al. reported that GPV and DuCV have synergistic effects on replication and pathogenicity in mixed-infection ducks. Mixed infection with GPV and DuCV can significantly inhibit the growth and development of young ducks and cause immune organ atrophy, pallor, and liver necrosis [20]. In addition, ducks infected with DuCV are more prone to bacterial infections, and the severity of the subsequent mixed infection is greater than that of the individual bacterial infections alone [16].

DuCV shows tropism for all immune organs, but there are differences in organ tropism among the different strains and genotypes of DuCV [20]. Therefore, different prevention and control measures should be implemented depending

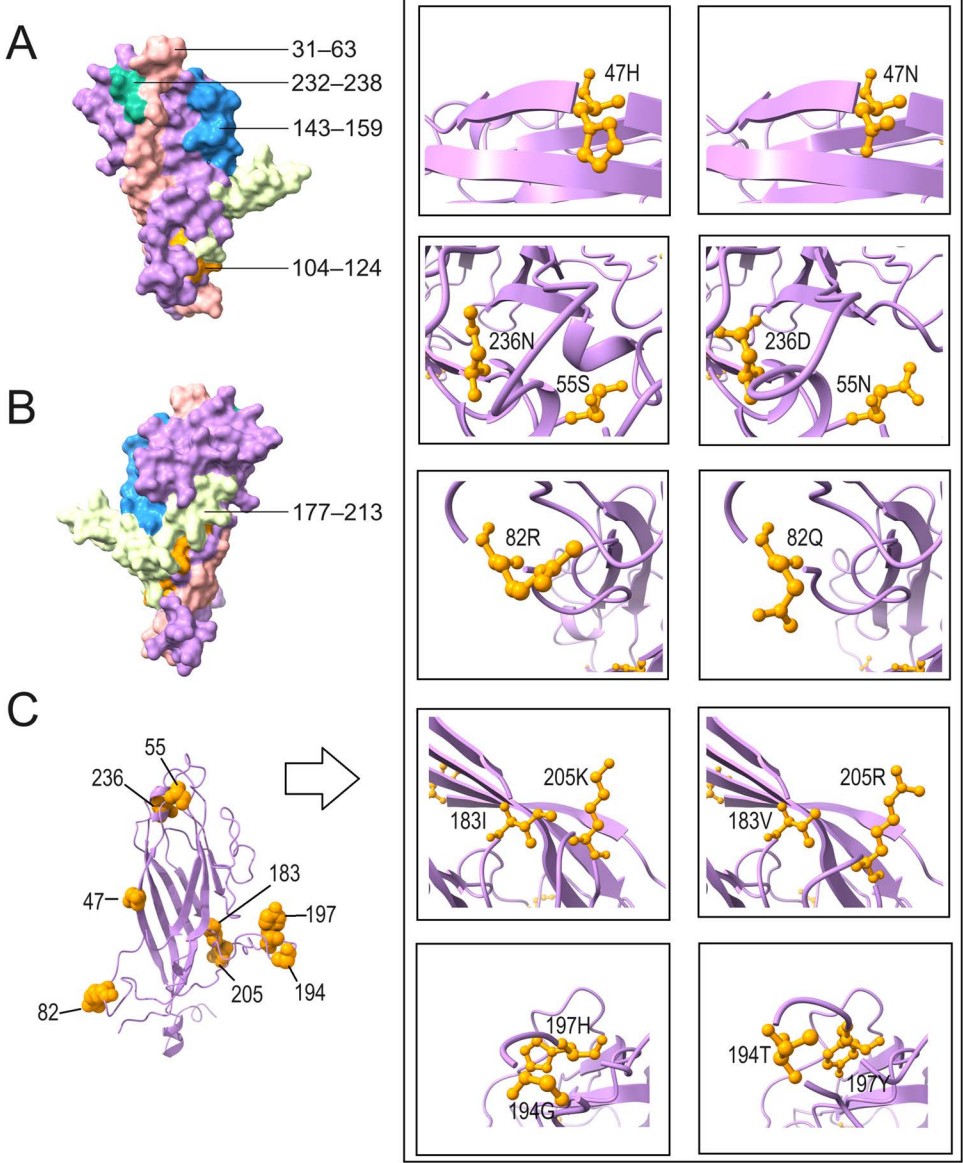

**Fig 7. Tertiary structure of the Cap protein predicted via Phyre2 (confidence in the model: 100%).** (A) and (B) Positions of the 5 variable regions; the reference sequences for both graphs are those of AY228555.1. The amino acids at positions 3–15 are not included the protein's tertiary structure. (C) Positions of some amino acids and the changes in protein shape due to gene mutations.

on the region and genotype of DuCV in question. According to genetic and evolutionary analyses of all of the DuCV genome sequences available in the NCBI database prior to 2023, a large number of DuCV genotypes were present in China before 2015, but after 2021, DuCV-1b and DuCV-2c began emerging as the main genotypes in circulation in China. The sequence analysis results in this study revealed two or more genotypes of DuCV in Shandong, Anhui, and Guangdong. In addition, some researchers have recently declared an epidemic of DuCV-1d in Anhui [21], which undoubtedly makes DuCV infection more difficult to prevent and control in this area. According to homologous recombination analysis, the LP0920-N2, FDD0930, C0720-Y7-2, and ZCY0822 sequences identified in this study were the major parents.

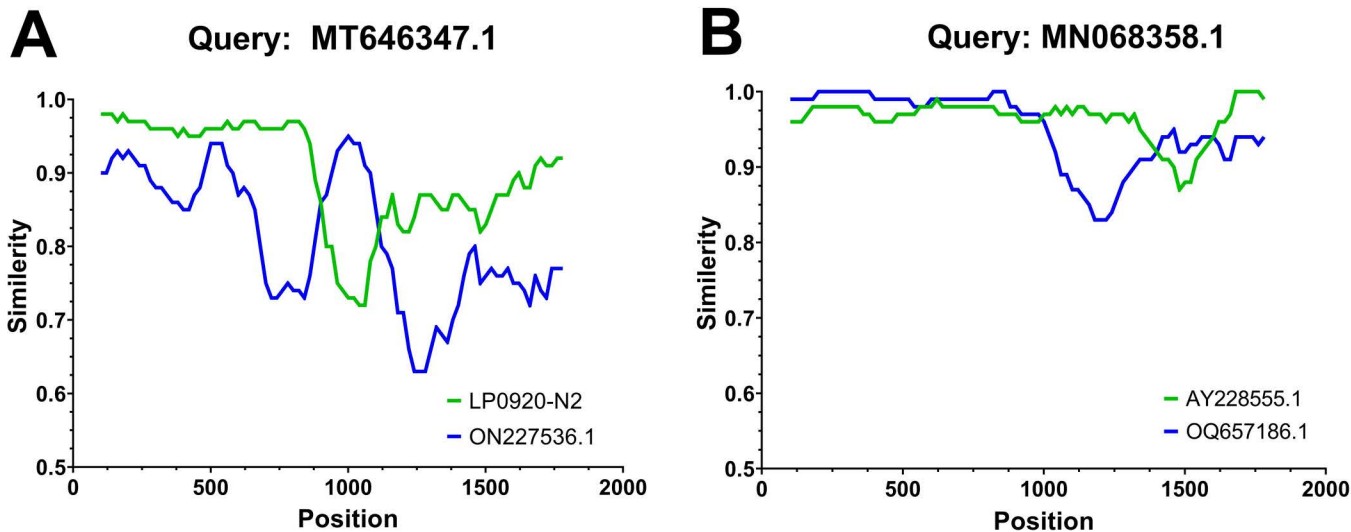

**Fig 8. Presentation of recombination events with Simplot.** (A) Recombination event 1. (B) Recombination event 7.

Interestingly, recombination of the German DuCV and the Chinese DuCV was observed in recombination event 7. DuCV can be transmitted across species, especially similar species such as wild ducks [3], as detected in the eastern coastal areas of China [8]. Eight wild bird migration routes have been defined across the world, 3 of which partially pass through regions of China, potentially leading to the international spread of DuCV [8]. Moreover, the most common recombination events involving DuCV have been detected in China, suggesting that the DuCVs circulating in China have wide genetic diversity.

The lack of an appropriate in vitro culture system [28,29] undoubtedly increases the difficulty of researching DuCV and developing a DuCV vaccine. A previous study demonstrated that DNA vaccines expressing the Cap protein of DuCV exhibited good immunogenicity [30]. However, the average distance between the amino acid sequences of the Cap proteins in the DuCV-1 and DuCV-2 groups was 0.133, and most of the amino acid mutations occurred at the potential epitopes of the Cap protein (S1 Fig), which suggests that the antigenicity of the two genotypes is substantially different. Therefore, cross-protection between different DuCV genotypes should be considered when developing a DuCV vaccine.

We also discovered an interesting phenomenon: in the DuCV-1b Cap protein, when the amino acids 47H, 183I, 197H, and 205K were present simultaneously, the amino acids 55N, 82Q, 106N, 107K, 194T, and 236D were not present, and vice versa. The Chinese DuCV-1b strains developed into two different clusters on the basis of the ORF2 (Fig 6). The average genetic distances of the nucleotide and amino acid sequences of these two clusters were 0.067 and 0.047, respectively, indicating the presence of differences between these two clusters. The amino acid residues at multiple sites, such as residues 55, 82, and 106, are located on the linear surface of B cells; therefore, changes at these sites may affect the antigenicity of the two clusters (S1 Fig). The prediction of glycosylation and phosphorylation sites at these positions indicates that the S55N mutation changes the phosphorylation site to a glycosylation site; similarly, changes in the amino acids at positions 106, 107, 194, and 197 also affect protein phosphorylation. Phosphorylation often causes changes in protein conformation [31], whereas changes to N-glycosylation sites play important roles in the replication and virulence of the influenza virus [32]. Studies have shown that the lack of N-glycosylation sites in the PCV2 Cap protein improves the specific immune response [33]. Therefore, these changes in these sites may also affect the antigenicity or immunogenicity of DuCV.

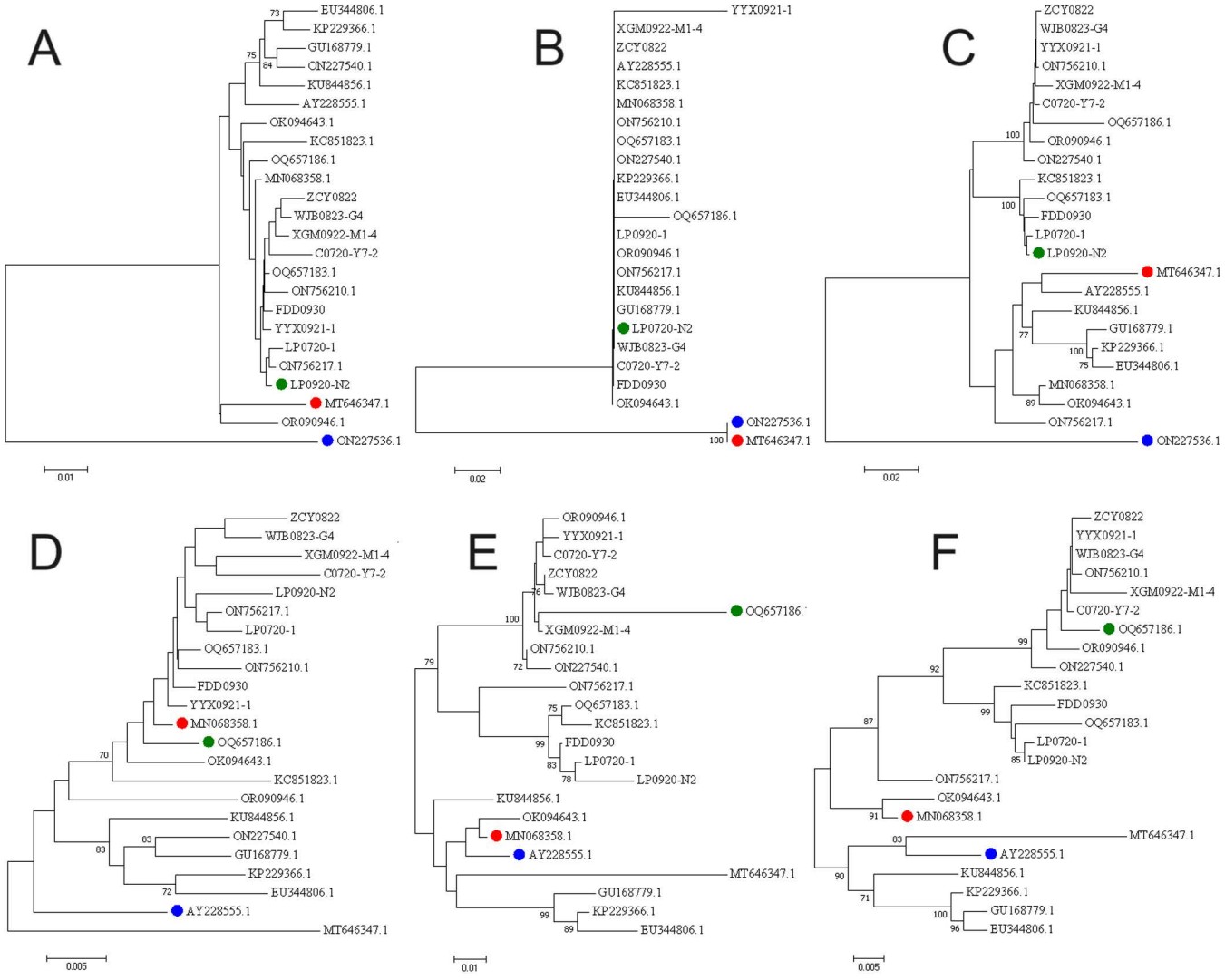

**Fig 9. Phylogenetic trees illustrating the potential recombination events shown in Fig 8. The nucleotide positions are consistent with the breakpoint positions in Table 3, representing the aligned positions following sequence alignment, not the original sequence positions.** For event 1 (A–C), the nucleotide sequence positions are 1–943, 944–1084, and 1085–1903; for event 7 (D–F), the nucleotide sequence positions are 1–979, 980–1402, and 1403–1903. The breakpoint locations represent the boundaries of the strongest recombination signal but are not necessarily close to the locations of the failures that occurred in the original recombination event.

## Materials and methods

### Sample collection and nucleic acid extraction

All tissue samples in this study were collected from ducks across 17 provinces in China by marketing managers and stored by the Group Testing Center of Shandong Sinder Technology Co., Ltd. (Shandong, China). The specific information of the detected samples is summarized in S2 Table. The collection of samples from the ducks was approved by the Experimental Animal Management and Ethics Committee of Shandong Sinder Technology Co., Ltd. (QXRZ: 2021–007); no other animal experiments were conducted. The samples were processed as 20% homogenates in phosphate-buffered saline. After grinding and subsequent centrifugation at 8000 × g at 4°C for 10 min, the clarified supernatant of each sample

was subjected to viral genome DNA extraction with an automatic nucleic acid extractor (Vazyme, Jiangsu, China). The viral DNA was stored at -80°C for subsequent research.

## Sample testing

All samples were simultaneously tested for DuCV and other pathogens requested by the marketing managers. The primers for DuCV F/R, PM F/R, and Staphylococcus F/R for the detection of DuCV, *Pasteurella multocida*, and *Staphylococcus,* respectively, are shown in Table 4. The remaining test items (novel goose parvovirus, Muscovy duck parvovirus, *Riemerella anatipestifer*, duck astrovirus, duck hepatitis A virus, duck reovirus, novel duck reovirus, avian influenza virus, duck Tembusu virus, duck enteritis virus, *Clostridium perfringens*, fowl adenovirus, or Salmonella) were purchased from Shandong Xinda Gene Technology Co., Ltd. (Weifang, China).

## Primer design for DuCV genome amplification`

A total of 53 DuCV genomes were downloaded from the NCBI database. The software MEGA 7 [34] was used for alignment and to determine the highly conserved regions of the DuCV genomes. Three pairs of primers were designed with Oligo 7 [35] to amplify the entire coding region of DuCV; BLAST analysis revealed no obvious nonspecific sequence similarities (Table 4). All the primers used in this study were purified via high-performance liquid chromatography (HPLC) and synthesized by Beijing Tsingke Biotech Co., Ltd. (Qingdao, China).

## PCR amplification and sequencing

PCR was performed via a commercial kit (GenStar, Beijing, China) in a 50 μL reaction volume. The reaction mixtures were composed of 25 μL of 2×Taq PCR StarMix, 16 μL of ddH$_2$O, 5 μL of viral DNA, and 2 μL each of the forward and reverse primers (each 10 μM). The reaction procedure was as follows: initial denaturation at 94°C for 4 min; 30 cycles of denaturation at 94°C for 30 s, annealing at 55°C for 30 s and extension at 72°C for 1 min; and final extension at 72°C for 5 min. After the reaction, the products were identified through 1% agarose gel electrophoresis and sequenced by Beijing Tsingke Biotech Co., Ltd. (Qingdao, China).

Table 4. PCR primers for detection and DuCV genome amplification.

| Primer names | Primer sequence (5' to 3') | Primer location[a] |
|---|---|---|
| DuCV F | CATGCCCATGCCGTAAT | 1259–1275 |
| DuCV R | TCAGAAGACGAAGGCTACG | 1845–1863 |
| DuCV F1 | AAACGGCGCTTGTACTCCGTAC | 3–21 |
| DuCV R1 | CTGGGCGGGTTCATACTTGTC | 874–894 |
| DuCV F2 | CATGGACGACTTTTATGGTTGG | 660–681 |
| DuCV R2 | CAACATTTACCAGAARAGCAARAC | 1518–1541 |
| DuCV F3 | CATGCCCATGCCGTAAT | 1259–1275 |
| DuCV R3 | GCGCTTGTGCGGTCTT | 1967–1982 |
| PM F | GGAAATGGCATTATTTTATGGC | |
| PM R | ACTTTTTGTTTCATTTGGACTGACAC | |
| Staphylococcus F | GCGATTGATGGTGATACGGTT | |
| Staphylococcus R | AGCCAAGCCTTGACGAACTAAAGC | |

[a]DuCV reference sequence: OK094643.1.

## Genome sequence analysis

The reference DuCV genome sequences of the different genotypes prior to 2023 were downloaded from the NCBI database for analysis [1,21]. The SeqMan software package from DNAStar [36] was used for sequence assembly, and MegAlign was used for gene identity analysis of the whole coding region sequence (via the Clustal W method).

A phylogenetic tree was constructed via the neighbor–joining method in MAGE-7 and inspected via 1,000 bootstrap iterations, and iTOL [37] was used to visualize the phylogenetic trees. The whole coding region sequence was deduced from the amino acid sequence, and the potential B-cell epitopes in the ORF2 region were predicted with BepiPred-3.0 [38]. The online software programs YinOYang1.2 (https://services.healthtech.dtu.dk/services/YinOYang-1.2/), NetNGlyc1.0 (https://services.healthtech.dtu.dk/services/NetNGlyc-1.0/), and NetPhos3.1 (https://services.healthtech.dtu.dk/services/NetPhos-3.1/) were used to analyze the O-glycosylation, N-glycosylation, and phosphorylation sites of the Cap protein of DuCV-1b, respectively. Phyre2 [39] online software was used to predict the tertiary structure of the Cap protein, which was subsequently visualized and edited with UCSF ChimeraX [40] software. RDP4 [41] software was used for homologous recombination analysis of all the sequences. Simplot [42] software was used to visualize the recombination events.

## Supporting information

**S1 Fig. Predicted B-cell epitopes located within the ORF2 region of Duck circovirus (GenBank accession: AY228555.1).** The BepiPred-3.0 online forecasting program was used with the default settings. The protein ID of the ORF2 protein of AY228555.1 is AAP69227.1.
(PNG)

**S1 Table. The information of Duck circoviruses sequenced in this study.**
(DOCX)

**S2 Table. Summary of sample information for this study.**
(DOCX)

## Author contributions

**Conceptualization:** Chunguo Liu.

**Data curation:** Peidong Li, Fuyou Zhang, Chunyang Bao, Hongmei Liu, Kai Yu, Hao Zhu, Xue Wang, Ke Shen.

**Investigation:** Peidong Li, Chunyang Bao.

**Methodology:** Peidong Li.

**Resources:** Zhaoyang Li.

**Supervision:** Qingqing Song, Chunguo Liu.

**Writing – original draft:** Peidong Li, Fuyou Zhang.

**Writing – review & editing:** Tianyao Yang, Chunguo Liu.

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
