## [Decision Letter · Decision Letter 0]

19 Jan 2025

PONE-D-24-53142Epidemiological investigation and genetic evolution analysis of duck circovirus in China, 2022PLOS ONE

Dear Dr. Liu,

Thank you for submitting your manuscript to PLOS ONE. After careful consideration, we feel that it has merit but does not fully meet PLOS ONE’s publication criteria as it currently stands. Therefore, we invite you to submit a revised version of the manuscript that addresses the points raised during the review process.

We look forward to receiving your revised manuscript.

Kind regards,

Li Xing

Academic Editor

PLOS ONE

Journal Requirements:

“This study was supported by the Shandong Sinder Technology Co., Ltd. (Project Number: JC20230313001).”

4. Thank you for stating the following in the Financial Disclosure section: 

“This study was supported by the Shandong Sinder Technology Co., Ltd. (Project Number: JC20230313001).”

We note that you received funding from a commercial source: Shandong Sinder Technology Co., Ltd.

5. In the online submission form you indicate that your data is not available for proprietary reasons and have provided a contact point for accessing this data. Please note that your current contact point is a co-author on this manuscript. According to our Data Policy, the contact point must not be an author on the manuscript and must be an institutional contact, ideally not an individual. Please revise your data statement to a non-author institutional point of contact, such as a data access or ethics committee, and send this to us via return email. Please also include contact information for the third party organization, and please include the full citation of where the data can be found.

7. We note that Figure 1 in your submission contain map images which may be copyrighted. All PLOS content is published under the Creative Commons Attribution License (CC BY 4.0), which means that the manuscript, images, and Supporting Information files will be freely available online, and any third party is permitted to access, download, copy, distribute, and use these materials in any way, even commercially, with proper attribution. For these reasons, we cannot publish previously copyrighted maps or satellite images created using proprietary data, such as Google software (Google Maps, Street View, and Earth). For more information, see our copyright guidelines: http://journals.plos.org/plosone/s/licenses-and-copyright.

1) You may seek permission from the original copyright holder of Figure 1 to publish the content specifically under the CC BY 4.0 license.  

2) If you are unable to obtain permission from the original copyright holder to publish these figures under the CC BY 4.0 license or if the copyright holder’s requirements are incompatible with the CC BY 4.0 license, please either i) remove the figure or ii) supply a replacement figure that complies with the CC BY 4.0 license. Please check copyright information on all replacement figures and update the figure caption with source information. If applicable, please specify in the figure caption text when a figure is similar but not identical to the original image and is therefore for illustrative purposes only.

8. Please upload a new copy of Figures 4, 6, and 9 as the detail is not clear. Please follow the link for more information

https://blogs.plos.org/plos/2019/06/looking-good-tips-for-creating-your-plos-figures-graphics/

https://blogs.plos.org/plos/2019/06/looking-good-tips-for-creating-your-plos-figures-graphics/

Reviewers' comments:

Reviewer's Responses to Questions

**Comments to the Author**

1. Is the manuscript technically sound, and do the data support the conclusions?

Reviewer #1: Partly

Reviewer #2: Yes

2. Has the statistical analysis been performed appropriately and rigorously? 

Reviewer #1: Yes

Reviewer #2: Yes

3. Have the authors made all data underlying the findings in their manuscript fully available?

Reviewer #1: No

Reviewer #2: Yes

4. Is the manuscript presented in an intelligible fashion and written in standard English?

Reviewer #1: Yes

Reviewer #2: Yes

5. Review Comments to the Author

Reviewer #1: This article provided some valuable information for understanding the evolution of DuCV, but a few modifications are necessary to improve the manuscript.

1. Authors should provide the approval number for animal experiments.

2. Authors should provide detailed information of the clinical samples, such as the collected time and place, duck breeds, number of samples tested, number of positive samples, etc. The detailed information should be provided in a table, and Fig 1 should be deleted.

3. English editing and academic writing should be improved.

4. “Shandong Province had the highest number (293) of positive samples.” If it is because of the large number of samples collected in Shandong, this conclusion makes no sense.

Reviewer #2: In this study, Li et al. investigated the epidemiology, genetic diversity, and evolutionary dynamics of Duck Circovirus (DuCV) in China. Through extensive sampling and sequencing, they identified two main circulating genotypes, DuCV-1b and DuCV-2c, and explored mixed infections with other pathogens. Additionally, the authors performed genetic recombination analyses to trace the possible origins of DuCV strains. Overall, this manuscript provides significant insights into the molecular epidemiology of DuCV.

Major points:

1. The authors detected co-infection with DuCV in the samples. Could you please provide a detailed description of how the co-infection was detected and include the primer sequences used?

2. The authors downloaded an unspecified number of full-genome sequences of DuCV from the NCBI database to construct the genetic evolutionary tree. However, detailed information, such as the isolation date, location, species, and GenBank IDs, is not provided. What criteria were used to select these full-genome sequences for analysis?

3. The quality of the figures is low. The authors should upload the figures in higher resolution. The phylogenetic trees, in particular, are difficult to read.

4. Line 347: Would constructing a Maximum Likelihood (ML) tree be more accurate?

5. Line 195: What are the "reference strains"? Is there any detailed information provided about them?

6. PLOS authors have the option to publish the peer review history of their article (what does this mean? ). If published, this will include your full peer review and any attached files.

**Do you want your identity to be public for this peer review?** For information about this choice, including consent withdrawal, please see our Privacy Policy .

Reviewer #1: No

Reviewer #2: No

---

## [Author Response · Author response to Decision Letter 1]

17 Mar 2025

Responses to the Academic Editor:

Journal Requirements:

Response 1: The authors reviewed and corrected the manuscript according to the "PLOS ONE style templates" requirements thoroughly.

Response 2: In this study, we only collected the swabs form the ducks and no other animal experiments were conducted. We also added the ethics statement in the "Sample collection and nucleic acid extraction" section.

“This study was supported by the Shandong Sinder Technology Co., Ltd. (Project Number: JC20230313001).”

Response 3: This study was supported by the Shandong Sinder Technology Co., Ltd. (Project Number: JC20230313001). The funders had no role in study design, data collection and analysis, decision to publish, or preparation of the manuscript.

4. Thank you for stating the following in the Financial Disclosure section:

“This study was supported by the Shandong Sinder Technology Co., Ltd. (Project Number: JC20230313001).”

We note that you received funding from a commercial source: Shandong Sinder Technology Co., Ltd.

Response 4: This study was supported by Shandong Sinder Technology Co., Ltd. (Grant No. JC20230313001). The funding source had no involvement in study design, data collection, analysis, interpretation, or the decision to submit this work for publication. This does not alter our adherence to PLOS ONE policies on sharing data and materials. We have fully disclosed these interests to PLOS and have established a conflict management plan to address any potential conflicts of interest that may arise from this funding arrangement. All authors declare that they have no known competing financial interests or personal relationships that could appeared to influence the work reported in this manuscript.

5. In the online submission form you indicate that your data is not available for proprietary reasons and have provided a contact point for accessing this data. Please note that your current contact point is a co-author on this manuscript. According to our Data Policy, the contact point must not be an author on the manuscript and must be an institutional contact, ideally not an individual. Please revise your data statement to a non-author institutional point of contact, such as a data access or ethics committee, and send this to us via return email. Please also include contact information for the third party organization, and please include the full citation of where the data can be found.

Response 5: We have uploaded all sample data as supplementary material labeled "S2 Table. Summary of sample information for this study" in the revised manuscript. Additionally, we have revised the Data Availability statement (The data presented in this study are openly available in GenBank under accession numbers OR387724 and OR387774. See S2 Table for the sample information.) in the submission system.

Response 6: We have added ethics statement in the "Sample collection and nucleic acid extraction" section.

7. We note that Figure 1 in your submission contain map images which may be copyrighted. All PLOS content is published under the Creative Commons Attribution License (CC BY 4.0), which means that the manuscript, images, and Supporting Information files will be freely available online, and any third party is permitted to access, download, copy, distribute, and use these materials in any way, even commercially, with proper attribution. For these reasons, we cannot publish previously copyrighted maps or satellite images created using proprietary data, such as Google software (Google Maps, Street View, and Earth). For more information, see our copyright guidelines: http://journals.plos.org/plosone/s/licenses-and-copyright.

1) You may seek permission from the original copyright holder of Figure 1 to publish the content specifically under the CC BY 4.0 license.

2) If you are unable to obtain permission from the original copyright holder to publish these figures under the CC BY 4.0 license or if the copyright holder’s requirements are incompatible with the CC BY 4.0 license, please either i) remove the figure or ii) supply a replacement figure that complies with the CC BY 4.0 license. Please check copyright information on all replacement figures and update the figure caption with source information. If applicable, please specify in the figure caption text when a figure is similar but not identical to the original image and is therefore for illustrative purposes only.

Response 7: The map images used in this study were obtained from the Resource and Environmental Science Data Platform (https://www.resdc.cn). The authors contacted the website administrator, who confirmed that this is an open-access platform providing freely available data for any purpose, including commercial use, with no copyright restrictions. Furthermore, the authors contacted the copyright holders of the map images and obtained explicit copyright authorization. The authorization document has been submitted as supplementary material.

8. Please upload a new copy of Figures 4, 6, and 9 as the detail is not clear. Please follow the link for more information

https://blogs.plos.org/plos/2019/06/looking-good-tips-for-creating-your-plos-figures-graphics/

Response 8: We have prepared a new clear copy of Figures 4, 6, and 9 using the provided link and uploaded to the submission system.

Responses to Reviewers' Comments:

Review Comments to the Author

Reviewer #1: This article provided some valuable information for understanding the evolution of DuCV, but a few modifications are necessary to improve the manuscript.

1. Authors should provide the approval number for animal experiments.

Response 1: We have added ethics statement in the "Sample collection and nucleic acid extraction" section.

2. Authors should provide detailed information of the clinical samples, such as the collected time and place, duck breeds, number of samples tested, number of positive samples, etc. The detailed information should be provided in a table, and Fig 1 should be deleted.

Response 2: The authors have uploaded all sample information as supplementary material labeled "S2 Table. Summary of sample information for this study". However, due to the extensive sample size, presenting the detailed data in tabular format within the main text is impractical. To enhance readability, we have retained Figure 1 in the main text while providing the complete dataset as supplementary material.

3. English editing and academic writing should be improved.

Response 3: The manuscript underwent professional language editing by "AJE: Academic Journal Editing Services " (https://www.aje.cn/) prior to submission. Following receipt of the editorial decision letter, the authors engaged AJE's academic editing team for an additional round of comprehensive editing. This rigorous process ensured adherence to the target journal's formatting guidelines and enhanced the manuscript's readability for the intended audience.

4. “Shandong Province had the highest number (293) of positive samples.” If it is because of the large number of samples collected in Shandong, this conclusion makes no sense.

Response 4: we have deleted the sentence of "Shandong Province had the highest number (293) of positive samples."

Reviewer #2: In this study, Li et al. investigated the epidemiology, genetic diversity, and evolutionary dynamics of Duck Circovirus (DuCV) in China. Through extensive sampling and sequencing, they identified two main circulating genotypes, DuCV-1b and DuCV-2c, and explored mixed infections with other pathogens. Additionally, the authors performed genetic recombination analyses to trace the possible origins of DuCV strains. Overall, this manuscript provides significant insights into the molecular epidemiology of DuCV.

Major points:

1. The authors detected co-infection with DuCV in the samples. Could you please provide a detailed description of how the co-infection was detected and include the primer sequences used?

Response 1: The authors have supplemented the description of other pathogen detection protocols in the "Sample testing" section within "Materials and methods", along with primers for conventional PCR assays and manufacturers of the commercial testing kits used.

2. The authors downloaded an unspecified number of full-genome sequences of DuCV from the NCBI database to construct the genetic evolutionary tree. However, detailed information, such as the isolation date, location, species, and GenBank IDs, is not provided. What criteria were used to select these full-genome sequences for analysis?

Response 2: To ensure data integrity and transparency, all full-length DuCV sequences used for phylogenetic tree construction in this study were obtained from the National Center for Biotechnology Information (NCBI), encompassing all available sequences deposited up to and including 2022. The GenBank accession numbers of the utilized sequences are explicitly annotated on the corresponding phylogenetic tree.

3. The quality of the figures is low. The authors should upload the figures in higher resolution. The phylogenetic trees, in particular, are difficult to read.

Response 3: We have prepared a new copy of each figure using the provided link (https://blogs.plos.org/plos/2019/06/looking-good-tips-for-creating-your-plos-figures-graphics/) by the editor and uploaded to the submission system.

4. Line 347: Would constructing a Maximum Likelihood (ML) tree be more accurate?

Response 4: We sincerely appreciate the reviewer' valuable suggestions. To validate the robustness of the phylogenetic analysis, we conducted a re-construction using Maximum Likelihood (ML) methods. The results confirmed that this methodological verification did not alter the genotypic classification data of DuCV or compromise the integrity of the study's core findings. So we have not made any change in the revised manuscript.

5. Line 195: What are the "reference strains"? Is there any detailed information provided about them?

Response 5: The "reference strains" referenced as the strains prior to 2023 downloaded from NCBI with full length genomic sequence, and we have edited in the revised manuscript. The accession numbers of all reference strains were showed in the Figure 4.

---

## [Decision Letter · Decision Letter 1]

6 Apr 2025

Epidemiological investigation and analysis of the genetic evolution of duck circovirus in China, 2022

PONE-D-24-53142R1

Dear Dr. Chunguo Liu,

We’re pleased to inform you that your manuscript has been judged scientifically suitable for publication and will be formally accepted for publication once it meets all outstanding technical requirements.

Kind regards,

Li Xing

Academic Editor

PLOS ONE

Additional Editor Comments (optional):

Reviewers' comments:

Reviewer's Responses to Questions

**Comments to the Author**

1. If the authors have adequately addressed your comments raised in a previous round of review and you feel that this manuscript is now acceptable for publication, you may indicate that here to bypass the “Comments to the Author” section, enter your conflict of interest statement in the “Confidential to Editor” section, and submit your "Accept" recommendation.

Reviewer #1: (No Response)

2. Is the manuscript technically sound, and do the data support the conclusions?

Reviewer #1: Yes

3. Has the statistical analysis been performed appropriately and rigorously? 

Reviewer #1: Yes

4. Have the authors made all data underlying the findings in their manuscript fully available?

Reviewer #1: Yes

5. Is the manuscript presented in an intelligible fashion and written in standard English?

Reviewer #1: Yes

6. Review Comments to the Author

Reviewer #1: In this study, the authors investigated the epidemiology, genetic diversity, and evolutionary dynamics of Duck Circovirus (DuCV) in China. Through extensive sampling and sequencing, they identified two main circulating genotypes, DuCV-1b and DuCV-2c, and explored mixed infections with other pathogens. Additionally, the authors performed genetic recombination analyses to trace the possible origins of DuCV strains. Overall, this manuscript provides significant insights into the molecular epidemiology of DuCV. The revised MS is OK.

7. PLOS authors have the option to publish the peer review history of their article (what does this mean? ). If published, this will include your full peer review and any attached files.

**Do you want your identity to be public for this peer review?** For information about this choice, including consent withdrawal, please see our Privacy Policy .

Reviewer #1: No

---

## [Editor Report · Acceptance letter]

PONE-D-24-53142R1

PLOS ONE

Dear Dr. Liu,

I'm pleased to inform you that your manuscript has been deemed suitable for publication in PLOS ONE. Congratulations! Your manuscript is now being handed over to our production team.

Kind regards,

on behalf of

Professor Li Xing

Academic Editor

PLOS ONE